# Comparison of pulmonary circulation parameters acquired by cardiovascular magnetic resonance with right heart catheterization and transthoracic echocardiography in patients with recent-onset dilated cardiomyopathy

Lukáš Opatřil[1‡], Mary Luz Mojica-Pisciotti[2‡], Roman Panovský[1*], Jan Máchal[3], Tomáš Holeček[4], Věra Feitová[5], Július Godava[6], Hana Poloczková[6], Vladimír Kincl[1], Michael Andrej[7], Jan Krejčí[6]

1 First Department of Internal Medicine and Cardioangiology, International Clinical Research Centre, St. Anne´s University Hospital, Faculty of Medicine, Masaryk University, Brno, Czech Republic, 2 International Clinical Research Centre, St. Anne´s University Hospital, Brno, Czech Republic, 3 Department of Pathophysiology, International Clinical Research Centre, St. Anne´s University Hospital, Faculty of Medicine, Masaryk University, Brno, Czech Republic, 4 Department of Medical Imaging, International Clinical Research Centre, St. Anne´s University Hospital, Department of Biomedical Engineering, Brno University of Technology, Brno, Czech Republic, 5 Department of Medical Imaging, International Clinical Research Centre, St. Anne´s University Hospital, Brno, Czech Republic, 6 First Department of Internal Medicine and Cardioangiology, St. Anne´s University Hospital, Faculty of Medicine, Masaryk University, Brno, Czech Republic, 7 Faculty of Medicine, Masaryk University, Brno, Czech Republic

‡ LO and MLMP contributed equally to this work and are joint first authors on this work.
* panovsky@fnusa.cz

## Abstract

### Introduction

Evaluating pulmonary circulation parameters (PCP) with cardiovascular magnetic resonance (CMR) is a relatively new approach with the potential for complex evaluation of the cardio-pulmonary system. Its impact might complement clinical assessment through right heart catheterization (RHC), the gold standard in evaluating pulmonary hypertension (PH) and hemodynamics, and transthoracic echocardiography (TTE). The study aims to examine the correlation between PCP and diastolic and systolic function, as well as PH, in patients with recent-onset dilated cardiomyopathy (RODCM).

### Methods

Eighty-four patients with RODCM were retrospectively included. All patients had a CMR examination, RHC (including pulmonary capillary wedge pressure (PCWP) and pulmonary vascular resistance (PVR)), and TTE. The pulmonary transit time (PTT), corrected pulmonary transit time (PTTc), systolic and diastolic function, and PH were

**Data availability statement:** We contacted the Data Protection Officer (DPO) of our hospital. With his agreement, we are now able to upload the data provided that it is not traceable and, therefore, anonymized. At our institution, we work with pseudonymized data; for example, including dates of examinations would not fulfill the conditions of our DPO. We uploaded a version approved by our DPO as supplementary material.

**Funding:** This study was supported by the AZV grant project under the Ministry of Health of the Czech Republic (Ministerstvo Zdravotnictví Ceské Republiky), grant nr. NU22-02-00418 and was written at Masaryk university as part of the project "New trends and the impact of comorbidities in the diagnosis, stratification, and therapy of cardiovascular diseases" number MUNI/A/1844/2025 with the support of the Specific University Research Grant, as provided by the Ministry of Education, Youth and Sports (Ministerstvo Školství, Mládeže a Tělovýchovy) of the Czech Republic in the year 2025. The funders had no role in study design, data collection and analysis, decision to publish, or preparation of the manuscript.

**Competing interests:** The authors have declared that no competing interests exist.

**Abbreviations:** BMI, body mass index; BSA, body surface area; b-TFE, balanced turbo field echo; CO, cardiac output; CI, cardiac index; CMR, cardiovascular magnetic resonance; CpcPH, combined post- and pre-capillary pulmonary hypertension; CVP, central venous pressure; DPG, diastolic pulmonary artery pressure; HR, heart rate; IpcPH, isolated post-capillary pulmonary hypertension; LV, left ventricle; LVEF, left ventricular ejection fraction; LVEDV, left ventricle end-diastolic volume; LVESV, left ventricle end-systolic volume; LVOT, left ventricular outflow tract; LVSV, left ventricular stroke volume; mPAP, mean pulmonary artery pressure; MRI, magnetic resonance imaging; PAPi, pulmonary artery pulsatility index; PBVI, pulmonary blood volume index; PCP, pulmonary circulation parameters; PCWP, pulmonary capillary wedge

assessed. Patients were divided into groups according to the PH and the diastolic function.

## Results

PTT and PTTc correlated with PCWP, cardiac index, PVR, and E/e'. Patients with a restrictive filling pattern showed significantly longer PTT. The receiver operating characteristic curves for PTT, PTTc, and PH were assessed with areas under the curve of 72.7% for PTT and 75.3% for PTTc, and cut-off values of 8.62 s (PTT) and 8.52 s (PTTc).

## Conclusion

To our knowledge, this is the first study focused on CMR-derived PCP in an RODCM group. Our findings show that PTT and PTTc are prolonged in patients with impaired systolic and diastolic function, and PH. Therefore, PCP might offer critical information to evaluate the cardio-pulmonary system comprehensively.

## Introduction

Pulmonary circulation parameters (PCP) are, in terms of cardiovascular magnetic resonance (CMR), a relatively new technique with the potential for complex evaluation of the cardio-pulmonary system as a whole [1–5]. While previously assessed by other modalities, such as radionuclide imaging [6], computed tomography [7], or contrast-enhanced transthoracic echocardiography (TTE) [8], CMR shows significant advantages and even allows retrospective analysis of these parameters [1,2].

Right heart catheterization (RHC) is the gold standard in evaluating pulmonary hypertension (PH) and hemodynamics [9]. In addition, compared to other PH evaluation methods, only RHC can differentiate between precapillary and postcapillary PH [9]. TTE, on the other hand, is the gold standard in diastolic function assessment [10,11].

A prolongation in PCP parameters has been documented by systolic dysfunction [3,12], diastolic dysfunction [2,13], and recently, PH as well [14,15]. Therefore, the data on the correlation between PCP, RHC, and TTE are crucial but limited. To our knowledge, no article including all three modalities and the PCP assessment in patients with recent-onset dilated cardiomyopathy (RODCM) has been published so far.

The study aims to provide such data and examine the correlation between PCP and diastolic and systolic function, as well as PH, in patients with RODCM, defined as newly diagnosed dilated cardiomyopathy with heart failure symptoms appearing in the last six months.

## Materials and methods

### Study design and population

This retrospective study was performed in accordance with the Declaration of Helsinki (2000) of the World Medical Association. The analysis used pseudonymized

pressure; PH, pulmonary hypertension; PTB, pulmonary transit beats; PTT, pulmonary transit time; PTTc, Bazett's formula corrected pulmonary transit time; PVR, pulmonary vascular resistance; RHC, right heart catheterization; RODCM, recent-onset dilated cardiomyopathy; ROC, receiver operating characteristic; ROIs, regions of interest; RVSWI, right ventricular stroke work index; RV, right ventricle; RVEF, right ventricle ejection fraction; RVSV, right ventricular systolic volume; SAX, short-axis; SI, signal intensity; SSFP, steady-state free precession; TE, echo time; TP, transpulmonary gradient; TR, repetition time; TTE, transthoracic echocardiography.

data from participants enrolled in a study approved by the Ethics Committee of the St. Anne´s University Hospital under No. 32V/2013, for which written informed consent was obtained. No minors were included. According to Czech legislation, no additional ethics approval was required for this retrospective analysis.

Eighty-four patients with RODCM already participating in a research project in our department were retrospectively included in this study. They had a CMR examination (including rest perfusion), RHC (including pulmonary capillary wedge pressure (PCWP) and pulmonary vascular resistance (PVR)), and TTE (including diastolic function assessment). CMR and RHC were performed within 0–8 days of each other. Patients were divided into groups according to their PH and diastolic function.

### CMR protocol

CMR studies were performed following our standard protocol [2,16] using a 1.5 T scanner (Ingenia, Philips Medical Systems, Best, The Netherlands) equipped with 5- and 32-element phased-array receiver coils, allowing for the use of parallel acquisition techniques in the supine position in repeated breath-holds. Functional imaging using balanced steady-state free precession (SSFP, b-TFE) cine sequences included four-chamber, two-chamber, and left ventricular outflow tract (LVOT) long-axis views, and a short-axis (SAX) stack from the cardiac base to the apex in the plane perpendicular to the LV long axis. LV functional and morphological parameters were calculated from the SAX stack using the summation-of-disc methods following the recommendations on post-processing evaluation from the Society for Cardiovascular Magnetic Resonance [17].

CMR first-pass contrast-enhanced myocardial perfusion images were acquired as described in previous studies [2]. Briefly, a b-TFE sequence acquired images in three SAX sections (basilar, midventricular, and apical) with these parameters: field of view $300 \times 300$ mm, reconstruction matrix 224, slice thickness 10 mm, acquisition voxel size $2.5 \times 2.5$ mm, time to repetition (TR) ≈ 2.2 ms, echo time (TE) ≈ 1.1 ms, flip angle 50°, SENSE factor 2.3, number of dynamics = 90, non-shared saturation prepulse. The images were acquired without breath-hold.

### Pulmonary circulation biomarkers analyses

Data obtained allowed the assessment of the PCP in accordance with our previous studies [2]. PCP parameters included pulmonary transit time (PTT) and pulmonary transit beats (PTB). The PTT and PTB values were estimated from SAX rest and stress first-pass perfusion images using a custom script developed in Python 3.7.16 (Python Software Foundation). A motion correction algorithm was applied to optimize image registration and avoid potential contamination from pixels in the blood pool. The registration accuracy was visually assessed. Regions of interest (ROIs) in the RV and the LV were manually traced in a sample image for the mid-ventricular slice and, in case of repetitive misregistration, in a basal one. The ROIs propagated throughout the stack of images, their average was computed, and signal intensity (SI) curves vs time were obtained. The algorithm identified the onset SI values; PTT was defined as

the difference between the LV and the RV onset time, and PTB was the number of frames between these times [2] (see Fig 1).

In addition, to reduce the effect of the heart rate (HR) on the results, corrected pulmonary transit time (PTTc) was calculated using Bazett's formula [18]. Likewise, PTTc (s) was defined as PTT (s)/√(RR interval/1 s), as shown previously [2,13].

## Right heart catheterization

The RHC was performed following the European Society of Cardiology 2022 guidelines [9]. A fluid-filled catheter and pressure transducer were used to measure the pressure. PH was defined as the mean pulmonary arterial pressure (mPAP) over 20 mmHg in rest conditions. Transpulmonary gradient (TPG) was calculated as the difference between mPAP and pulmonary capillary wedge pressure (PCWP). Cardiac output (CO) was determined using the thermodilution method and indexed to BSA as cardiac index (CI). Finally, the pulmonary vascular resistance (PVR) was calculated as (mPAP - PCWP)/ CO.

Precapillary PH was defined as mPAP > 20 mmHg, PCWP ≤ 15 mmHg and PVR > 2 WU; isolated post-capillary pulmonary hypertension (IpcPH) as mPAP > 20 mmHg, PCWP > 15 mmHg and PVR ≤ 2 WU; and combined postcapillary and precapillary pulmonary hypertension (CpcPH) as mPAP > 20 mmHg, PCWP > 15 mmHg and PVR > 2 WU [9].

In addition, the Pulmonary Artery Pulsatility Index (PAPi), as a clinically validated hemodynamic parameter used to assess RV function was calculated. It was defined as:

$$PAPi = \frac{\text{pulmonary artery systolic pressure } - \text{ pulmonary artery diastolic pressure}}{\text{right atrial pressure}}$$

## Transthoracic echocardiography

All patients underwent the standard TTE exam, and all examinations were performed by experienced cardiologists in our center and according to the guidelines.

Diastolic function was evaluated in accordance with established recommendations [19,20] based on Doppler echocardiography. Patients were divided into subgroups depending on the diastolic dysfunction grade (grade I to IV).

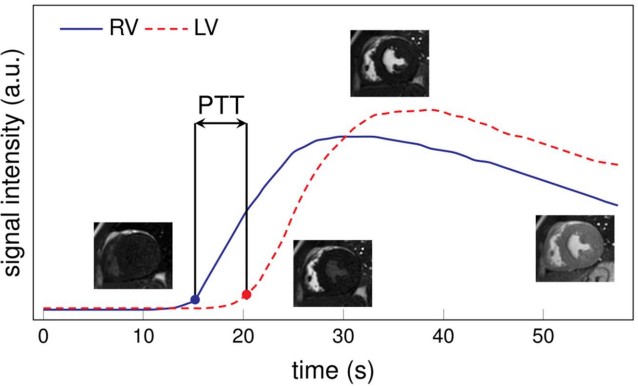

**Fig 1. Pulmonary transit time analysis.** Regions of interest (not shown) are manually traced in the right ventricle (RV) and the left ventricle (LV) to create the signal intensity (SI) vs. time curves. The pulmonary transit time (PTT) corresponds to the difference between the onsets, selected as the points where the signal surpasses 10% of the maximum values.

## Statistical analysis

PTT and PTB were correlated with echocardiographic and RHC markers using the Pearson correlation coefficient; logarithmic transformation was employed when needed to fit the normal distribution (including both PTB and PTT). The different patterns of diastolic dysfunction were compared using ANOVA with the Tukey post hoc test for unequal samples. The ability of PTT and PTTc to determine the presence of RHC-defined PH was expressed by the area under the receiver operating characteristic (ROC) curve, and a cut-off point was proposed based on the highest Youden index.

Intra- and inter-observer reproducibility were assessed using the intraclass correlation coefficient (ICC) (type C, two-way mixed-effects model with 95% confidence intervals (CI) from ten randomly selected cases. These cases were analyzed by two readers, one of whom repeated the analysis two weeks apart. The repeatability was classified as poor (<0.5), fair (0.50 to 0.75), good (0.75 to 0.90), and excellent (0.90 to 1). Statistica (version 14.0.0.15, TIBCO software) and R (version 3.6.1, R Foundation for Statistical Computing) were used for the analyses.

## Results

### Patient´s population

The mean age in our group was 50.5 ± 11.3 years, mean BMI 28.5 ± 5.4 kg/m2, and BSA 2.1 ± 5.4 m2. Patients had a reduced LV ejection fraction (LVEF) of 30.4 ± 11.2%, RV ejection fraction (RVEF) of 46.7 ± 15%, and an increased mPAP of 22.4 ± 8.6 mmHg. For a detailed list of basic clinical, CMR, RHC, and TTE parameters, see Table 1.

### Transthoracic echocardiography and right heart catheterization

TTE assessment of diastolic function was based on Doppler echocardiography of transmitral velocities of transmitral flow and tissue Doppler imaging of mitral annular velocity. Seventeen patients had normal diastolic function, 35 had impaired relaxation, 13 had pseudonormalization, and 19 had restrictive filling patterns. In addition, E/A and E/e´ were also entered as separately analyzed parameters.

According to the RHC, the mean mPAP of the population was 22.4 ± 8.6, PVR 1.84 ± 0.8, CVP 5.4 ± 3.6, PCWP 14.25 ± 7.64, and CI 2.36 ± 0.62. The patients were divided into subgroups according to the signs of PH: 47 had signs of PH (PH group), and 37 had no signs (non-PH group). Among the 47 patients with PH, 6 had isolated precapillary PH, 18 had IpcPH, and 22 had CpcPH. In one particular case, the patient had elevated mPAP over 20 mmHg, therefore signs of PH, but neither elevated PCWP nor PVR; thus, the patient could not be assigned to any category.

Of the six patients with precapillary PH, three were diagnosed with underlying lung disease at the time of the RHC; one patient was a smoker with a high likelihood of undetected lung pathology, and in two cases the cause remained unknown.

### Pulmonary circulation parameters

Intra-observer reproducibility was excellent for both PTB (ICC 0.992, 95% CI 0.970–0.998) and PTT (ICC 0.988, 95% CI 0.953–0.997). Likewise, inter-observer reproducibility was also excellent for PTB (ICC 0.982, 95% CI 0.934–0.996) and PTT (ICC 0.969, 95% CI 0.883–0.992).

All PTT, PTTc, and PTB had moderate correlations (ranging from approximately 0.37 to 0.61) with LVEF, PCWP, mean PAP, cardiac output, and E/A. All values are shown in Table 2.

In addition, only PTT and PTTc showed a moderate correlation with PVR and E/e'. Other parameters correlating with PCP included the peak systolic myocardial velocity (S') of the interventricular septum (IVS) assessed by pulsed-wave tissue Doppler imaging (TDI) and the central venous pressure (CVP). All correlations are shown in Table 2.

Patients with restrictive filling pattern (diastolic dysfunction grade III) showed significantly longer PTT compared to subgroups without diastolic dysfunction and with impaired relaxation (p = 0.01; p < 0.01 resp.) (see Fig 2).

**Table 1. Baseline clinical, TTE, CMR, and RHC parameters.**

| Parameters | Value | n |
|---|---|---|
| Number of patients | 84 | 84 |
| Age (years) | 50.5±11.3 | 84 |
| Gender (male), (%) | 76.2 | 64 |
| Body mass index (kg/m$^2$) | 28.5±5.4 | 84 |
| Body surface area (m$^2$) | 2.1±0.2 | 84 |
| Heart rate (BPM) | 77.7±17.1 | 84 |
| **TEE parameters** | | |
| LVEF – Estimation (%) | 24.4±9.3 | 84 |
| LVEF – Simpson (%) | 26.6±9.9 | 84 |
| E/A | 1.5±1.1 | 72 |
| E/e´ | 13.7±7.6 | 82 |
| TRPG (mmHg) | 22.7±13.4 | 84 |
| TDI – IVS S (m/s) | 0.05±0.02 | 82 |
| TDI – IVS E (m/s) | 0.07±0.13 | 82 |
| TDI – IVS A (m/s) | 0.07±0.04 | 71 |
| sPAP (mmHg) | 26.4±14.9 | 84 |
| **CMR parameters** | | |
| LVEF (%) | 30.4±11.2 | 84 |
| LVEDV (mL/m$^2$) | 254.6±98.2 | 84 |
| LVESV (mL/m$^2$) | 184.4±93.5 | 84 |
| PTB | 10.1±5.5 | 84 |
| PTT (s) | 8.10±3.56 | 84 |
| PTTc (s) | 9.19±4.41 | 84 |
| PBVI (mL/m$^2$) | 317.9±151.1 | 84 |
| **RHC parameters** | | |
| mPAP (mmHg) | 22.4±8.6 | 84 |
| PCWP (mmHg) | 14.3±7.6 | 84 |
| PVR (dyn-s/cm$^5$) | 1.8±0.8 | 84 |
| CVP (mmHg) | 5.4±3.6 | 84 |
| TP (mmHg) | 8.3±2.5 | 84 |
| DPG (mmHg) | 0.00±2.73 | 84 |
| PAPi | 6.4±5.6 | 84 |
| RVSWI (gm/m$^2$) | 6.3±2.4 | 84 |
| CI (L/min/m$^2$) | 2.4±0.6 | 84 |

Values are presented as mean ± SD unless otherwise indicated.

CI (Cardiac Index); CVP (Central Venous Pressure); DPG (Diastolic Pressure Gradient); LVEF (Left ventricle ejection fraction); LVEDV (Left ventricle end-diastolic volume); LVESV (Left ventricle end-systolic volume); mPAP (Mean Pulmonary Artery Pressure); PAPi (Pulmonary Artery Pulsatility Index); n (Number of values); PBVI (Pulmonary Blood Volume Index); PCWP (Pulmonary Capillary Wedge Pressure); PTB (Pulmonary Transit Beats); PTT (Pulmonary Transit Time); PTTc (Corrected Pulmonary Transit Time); PVR (Pulmonary Vascular Resistance); RVSWI (Right Ventricular Stroke Work Index); sPAP (Systolic Pulmonary Artery Pressure); TDI (Tissue Doppler Imaging), TP (Transpulmonary gradient); TRPG (Tricuspid Regurgitation Pressure Gradient),

**Table 2. Pulmonary circulation parameters correlations.**

| Parameter | Correlation | PTB | PTT | PTTc |
|---|---|---|---|---|
| LVEF | r | −0.55 | −0.48 | −053 |
| | P | < 0.001 | < 0.001 | < 0.001 |
| mPAP | r | 0.37 | 044 | 043 |
| | P | < 0.001 | < 0.001 | < 0.001 |
| PCWP | r | 0.40 | 048 | 047 |
| | P | < 0.001 | < 0.001 | < 0.001 |
| CVP | r | 0.24 | 032 | 027 |
| | P | 0.033 | < 0.001 | 0015 |
| CO | r | −0.41 | −049 | −047 |
| | P | < 0.001 | < 0.001 | < 0.001 |
| CI | r | −0.47 | −061 | −055 |
| | P | < 0.001 | < 0.001 | < 0.001 |
| PVR | r | 0.29 | 043 | 038 |
| | P | 0.104 | 0011 | 0030 |
| TDI – IVS | r | −0.36 | −0.42 | −0.40 |
| | P | 0.040 | 0.015 | 0.020 |

CVP (Central Venous Pressure); LVEF (Left ventricle ejection fraction); mPAP (mean Pulmonary artery pressure); PCWP (Pulmonary Capillary Wedge Pressure); PTB (Pulmonary Transit Beats); PTT (Pulmonary Transit Time); PTTc (Corrected Pulmonary Transit Time); PVR (Pulmonary Vascular Resistance); r (correlation coefficient); TDI (Tissue Doppler Imaging).

PTB, PTT, PTTc, CVP, CO, CI and PVR denote variables with log-normal distribution; logarithmic transformation was applied in these cases.

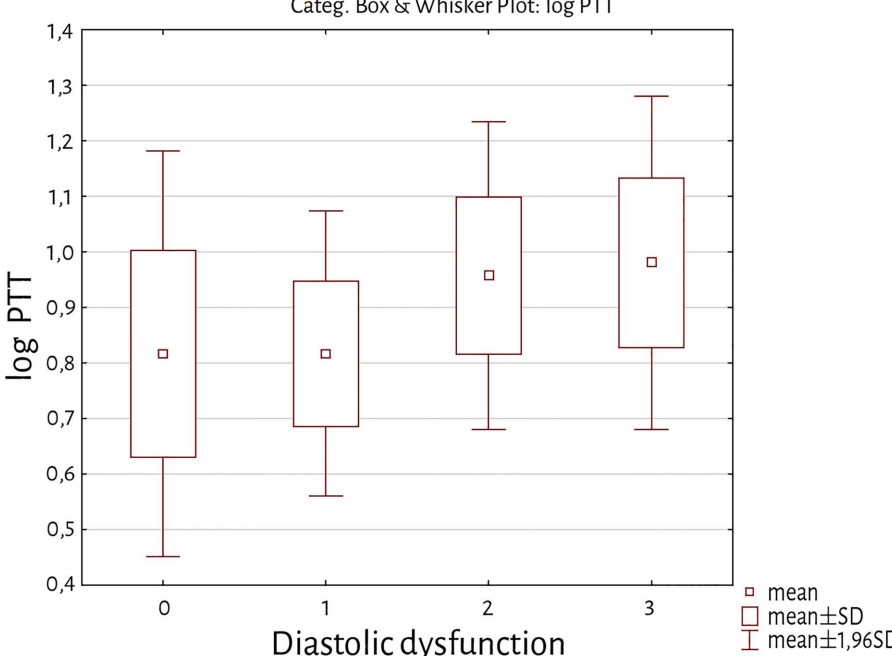

**Fig 2. Box-plot comparison of subgroups with different diastolic function.** Patients with diastolic dysfunction grade III (restrictive filling pattern, group 3) showed significantly longer PTT compared to subgroups without diastolic dysfunction (group 0) and with diastolic dysfunction grade II (impaired relaxation, group 2).

Finally, the PTT, PTTc, and PH ROC curves were assessed. In the case of PTT, the area under the curve was 72.7%, and the Youden index-based cut-off value was 8.62 s, with a sensitivity of 53.2% and a specificity of 86.5%. For PTTc, the area under the curve was 75.3%, and the Youden index-based cut-off value was 8.52 s, with a sensitivity of 70.2% and a specificity of 81.5% (see Fig 3 and Fig 4).

## Discussion

To our knowledge, this is the first study aimed to explore PCP parameters in RODCM patients involving three modalities (CMR, TTE, RHC) as gold standards for assessing systolic, diastolic function, and PH. Our findings suggest that the prolongation of PCP strongly correlates with parameters measured by CMR, RHC, and TTE, such as PVR, E/e´, LVEF, PCWP, CI, and grade III diastolic dysfunction, demonstrating the ability of PTT to differentiate patients with PH using CMR. Similarly to what we have described previously [1], although still limited, current data show a strong predictive value for the prolongation of PCP across various cardiac diagnoses, making this dynamically developing topic attractive for clinical applications today. From a pathophysiological point of view, longer times correspond to impairments of either the pumping function of the heart (systolic for both ventricles, diastolic for the LV) or the lungs'response (foremost PH). This behaviour implies that these parameters have a unique potential to assess the cardiopulmonary system. To date, no CMR-based method reliably estimates markers of PH, and even the assessment of diastolic function remains highly limited. This approach, combined with the simplicity of their acquisition and

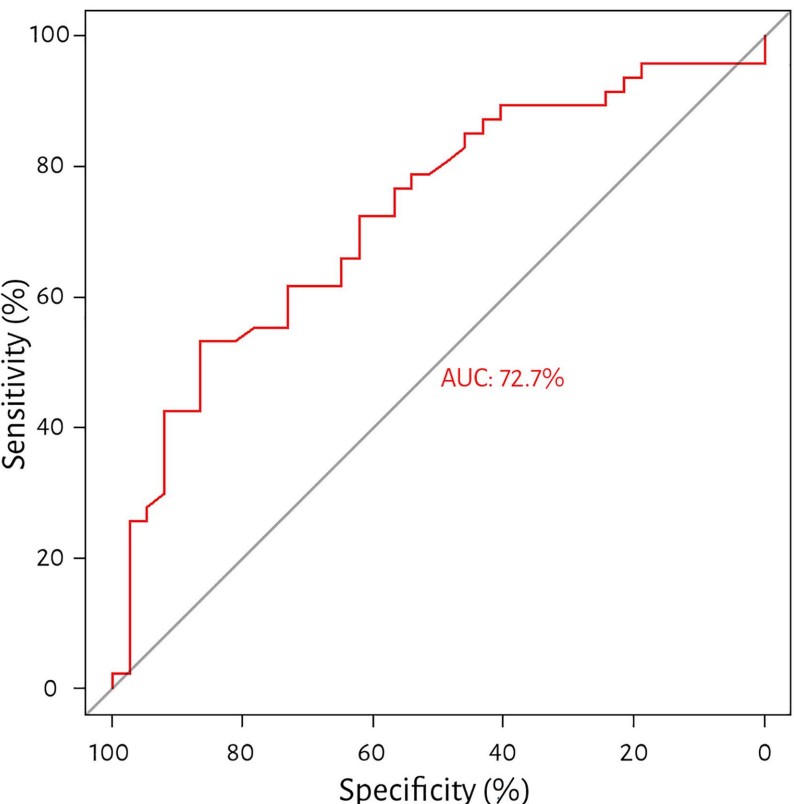

**Fig 3. Receiver operating characteristics for pulmonary transit time and pulmonary hypertension.** Receiver operating characteristics demonstrating the ability of pulmonary transit time (PTT) to determine pulmonary hypertension with a threshold of 8.62, sensitivity of 53.2%, specificity of 86.5%, and an area under the curve of 72.7%.

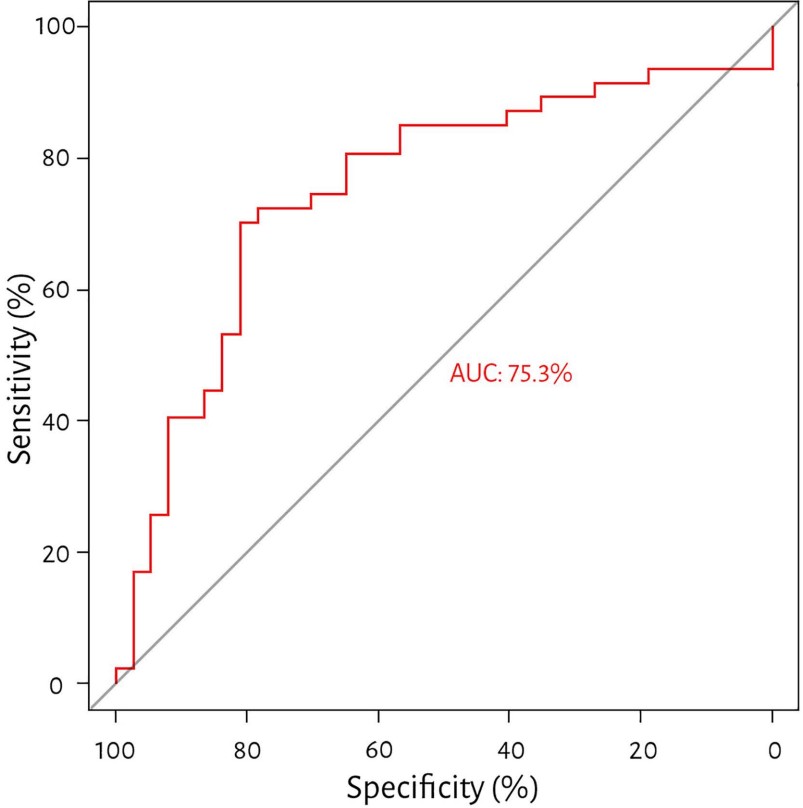

**Fig 4. Receiver operating characteristics for corrected pulmonary transit time and pulmonary hypertension.** Receiver operating characteristics demonstrating the ability of corrected pulmonary transit time (PTTc) to determine pulmonary hypertension with a threshold of 8.52, sensitivity of 70.2%, specificity of 81.1%, and an area under the curve of 75.3%.

the growing body of evidence supporting their predictive value, makes them a highly promising marker for the future of integrated cardiopulmonary diagnostics.

Expanding the available PCP data with correlations to RHC, which are currently severely limited, our results further demonstrate the potential of these fairly new parameters to contribute significantly to the assessment of pulmonary circulation as a whole, giving them great potential to complement existing diagnostic tools.

In addition, this is the first study to focus on CMR-based PCP assessment in patients with RODCM. The time between CMR and RHC in our cohort was only 8 days at most, while it was up to 30 days in previous works [14,15]. The mean time delay between CMR and RHC was 1.8±2.1 days and in the case of CMR and TTE 4.7±12.1 days. Since hemodynamics can be influenced by many different things, such as congestion, this approach reduces potential bias and ensures methodological consistency.

Finally, we also found that PTB showed less correlation with the studied parameters, while PTT and PTTc are comparable in their diagnostic ability, making them preferable for assessment, as they offer a more optimal option for differentiating PH using CMR.

This study has some limitations. Foremost, it is a retrospective single-centre study. Another limitation is our patient selection, RODCM. In them, isolated precapillary PH is rare; therefore, data on comparing precapillary with other types of PH are severely limited. One of the objectives of this study was to determine whether PCP can distinguish patients with isolated pre- or postcapillary PH. Although these parameters (particularly PTT and PTTc) correlate with PCWP, they also

show a weaker correlation with PVR. In our dataset, the cohort consisted of patients with newly diagnosed HF referred to our center, and as expected, the largest subgroup was those with CpcPH. In contrast, the number of patients with isolated precapillary PH was very limited. Partly because of this, we were unfortunately unable to differentiate isolated pre- or post-capillary PH using PCP alone reliably. It would also be invaluable to study prognosis in RODCM patients based on PCP, but we were not able to do so based on our data. However, we plan to carry out such a study in the future.

## Conclusion

To our knowledge, this is the first study focused on CMR-derived PCP in a RODCM group, including three diagnostic modalities as gold standards for assessing systolic function, diastolic function, and PH. Our findings show that PTT and PTTc are prolonged with impaired systolic and diastolic function as well as PH. Therefore, PTT and PTTc might offer critical information for a comprehensive evaluation of the cardio-pulmonary system as a whole.

## Supporting information

**S1 Table. PH subgroups.**
(XLSX)

**S2 Table. Source data file.**
(XLSX)

## Author contributions

**Conceptualization:** Lukáš Opatřil, Mary Luz Mojica-Pisciotti, Roman Panovský, Julius Godava, Vladimír Kincl, Michael Andrej, Jan Krejčí.

**Data curation:** Lukáš Opatřil, Mary Luz Mojica-Pisciotti, Jan Máchal, Věra Feitová, Hana Poloczková, Michael Andrej.

**Formal analysis:** Lukáš Opatřil, Věra Feitová, Julius Godava, Hana Poloczková, Vladimír Kincl, Jan Krejčí.

**Funding acquisition:** Roman Panovský, Hana Poloczková, Vladimír Kincl, Jan Krejčí.

**Investigation:** Lukáš Opatřil, Roman Panovský, Tomáš Holeček, Věra Feitová, Julius Godava, Vladimír Kincl, Michael Andrej, Jan Krejčí.

**Methodology:** Lukáš Opatřil, Mary Luz Mojica-Pisciotti, Roman Panovský, Jan Máchal, Tomáš Holeček, Věra Feitová, Jan Krejčí.

**Project administration:** Lukáš Opatřil, Tomáš Holeček.

**Software:** Mary Luz Mojica-Pisciotti, Jan Máchal.

**Supervision:** Roman Panovský, Hana Poloczková, Jan Krejčí.

**Visualization:** Věra Feitová.

**Writing – original draft:** Lukáš Opatřil, Mary Luz Mojica-Pisciotti.

**Writing – review & editing:** Lukáš Opatřil, Mary Luz Mojica-Pisciotti, Roman Panovský, Jan Máchal, Tomáš Holeček, Věra Feitová, Julius Godava, Hana Poloczková, Vladimír Kincl, Michael Andrej, Jan Krejčí.

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
