## [Decision Letter · Decision Letter 0]

27 Oct 2025

Dear Dr. Panovský,

Thank you for submitting your manuscript to PLOS ONE. After careful consideration, we feel that it has merit but does not fully meet PLOS ONE’s publication criteria as it currently stands. Therefore, we invite you to submit a revised version of the manuscript that addresses the points raised during the review process.

We look forward to receiving your revised manuscript.

Kind regards,

Wolfgang Rudolf Bauer, M.D., Ph.D.

Academic Editor

PLOS ONE

Journal Requirements:

4. In the online submission form, you indicated that data cannot be shared publicly - eventough it is pseudoanonymized, it still contains potentially confident data about our patients and therefore we cannot share it publicly. Data are available from the Internationa Clinical Research Center Institutional Data Access (contact via corresponding author - ass. prof. Panovský) for researchers who meet the criteria for access to confidential data.

6. Thank you for stating the following financial disclosure:

Supported by the project National Institute for Research of Metabolic and Cardiovascular Diseases (Programme EXCELES, ID Project No. LX22NPO5104) – Funded by the European Union – Next Generation EU and by AZV grant project undert Ministry of Health of the Czech Republic, grant nr. NU22-02-00418.

Reviewers' comments:

Reviewer's Responses to Questions

**Comments to the Author**

1. Is the manuscript technically sound, and do the data support the conclusions?

Reviewer #1: Yes

Reviewer #2: Yes

2. Has the statistical analysis been performed appropriately and rigorously?

Reviewer #1: Yes

Reviewer #2: Yes

3. Have the authors made all data underlying the findings in their manuscript fully available?

Reviewer #1: Yes

Reviewer #2: Yes

4. Is the manuscript presented in an intelligible fashion and written in standard English?

Reviewer #1: Yes

Reviewer #2: Yes

Reviewer #1: systolic and diastolic function and pulmonary hypertension in 84 patients with recent-onset dilated cardiomyopathy. All patients had a CMR examination, RHC (including pulmonary capillary wedge pressure (PCWP) and pulmonary vascular resistance 42 (PVR)), and TTE. The pulmonary transit time (PTT) and corrected pulmonary transit time 43 (PTTc) correlated with PCWP, cardiac index, PVR, and E/e’. ROC AUC was of 72.7% for PTT and 75.3% for PTTc, with cut-off values of 8.62 s (PTT) and 8.52 s (PTTc). The authors conclude that CMR-derived PCP might offer critical information to evaluate the cardio-pulmonary system comprehensively in patients with DCM.

The manuscript deals with an interesting topic and is based on sound methodology,

There are some points the authors might wish to address to further improve the manuscript.

1) Patients were divided in groups according to their PH and diastolic function. What was the respective rationale? Please specify.

2) Please provide quality assurance measures for CMR parameters (inter- and intra-observer agreement)

3) Is the software and algorithms used in this project freely available/transferable to other centers/machines?

4) What was the time delay between the different assessments (CMR, Echo, RHC)?

5) Table 1: gender: the n is missing

6) Table 1: “TEE” might rather the “TTE”

7) Table 1: the number of decimals should be equal for all numbers. One decimal might be sufficient

8) Table 1: for consistency, please also provide TTE LVEF and TR Vmax or TRmaxPG

9) Since you report a correlation with IVS, please provide the respective value in table 1

10) What parameters was the diagnosis of RODCM based on? Signs/Symptoms, NT-proBNP, Imaging markers??

Reviewer #2: General comments

The manuscript presents a retrospective cohort study analyzing pulmonary circulation parameters (Pulmonary Transit Beats [PTB], Pulmonary Transit Time [PTT], and corrected PTT [PTTc]) derived from cardiovascular magnetic resonance (CMR) imaging in comparison with right heart catheterization (RHC) and echocardiographic parameters, in 84 patients with recent-onset dilated cardiomyopathy. The authors demonstrate that PTT and PTTc correlate with systolic and diastolic cardiac function as well as hemodynamic indices from RHC, and that these metrics predict pulmonary hypertension (PH).

Specific comments

1. Table 1 Formatting

- Please avoid reporting unnecessarily precise values, e.g., heart rate of 77.67 ± 17.08. Rounding should be applied where appropriate.

- Add a column indicating the number of available data points for each parameter. If all values are complete, mention this briefly in the Methods section.

- Present mean and systolic pulmonary artery pressure as well as cardiac index (CI) in Table 1. The data for mPAP and CI are already presented on page 11, line 183.

- Define “pulmonary artery pressure index” in the Methods section.

- DPG refers to “diastolic pressure gradient,” not “Diastolic Pulmonary Artery Pressure” (page 11, line 174). DPG is the difference between pulmonary artery diastolic pressure and pulmonary capillary wedge pressure. Please provide this definition in the Methods.

2. Subgroup Division and Classification (Page 11, lines 184–187)

The numbers cited for PH subgroups do not add up to the total of 84 patients. Please verify and correct the classification.

- Remove the 28 “precapillary” entry; do not sum isolated precapillary patients with combined post- and precapillary PH.

- One patient remains unclassified; please clarify.

The correct breakdown should be:

- 37 patients without PH

- 47 patients with PH

--6 with precapillary PH.

--40 with postcapillary PH

---18 isolated postcapillary PH (IpcPH)

---22 combined post- and precapillary PH (CpcPH)

3. Please provide a more detailed characterization of the patients classified as having precapillary pulmonary hypertension. Specifically, did these individuals have underlying lung disease or other identifiable etiologies for precapillary PH? Alternatively, did they exhibit features typical of postcapillary PH, apart from the absence of PAWP elevation? Additionally, please report the average pulmonary artery wedge pressure (PAWP) for this subgroup.

4. Can you provide a table divided by patients with no PH, pcPH and prePH illustrating all CMR and RHC parameters?

5. Completeness of Table 1 (Page 11, line 188)

Table 1 does not present a complete list of evaluated parameters. Please ensure all relevant variables are included or adjust the related text accordingly.

6. Discussion

- The first two discussion paragraphs are repetitive. Please streamline and avoid using nearly identical sentences.

- The discussion is brief. Please elaborate on the clinical advantage of assessing PTB, PTT and PTTc by CMR compared to standard echocardiographic evaluation of PH.

State whether PTT or PTTc can predict increased pulmonary artery wedge pressure (PAWP), as this would markedly increase the clinical utility of your results.

**Do you want your identity to be public for this peer review?** For information about this choice, including consent withdrawal, please see our Privacy Policy

Reviewer #1: No

Reviewer #2: No

---

## [Author Response · Author response to Decision Letter 1]

16 Dec 2025

Review Comments to the Author

Reviewer #1:

systolic and diastolic function and pulmonary hypertension in 84 patients with recent-onset dilated cardiomyopathy. All patients had a CMR examination, RHC (including pulmonary capillary wedge pressure (PCWP) and pulmonary vascular resistance 42 (PVR)), and TTE. The pulmonary transit time (PTT) and corrected pulmonary transit time 43 (PTTc) correlated with PCWP, cardiac index, PVR, and E/e’. ROC AUC was of 72.7% for PTT and 75.3% for PTTc, with cut-off values of 8.62 s (PTT) and 8.52 s (PTTc). The authors conclude that CMR-derived PCP might offer critical information to evaluate the cardio-pulmonary system comprehensively in patients with DCM.

The manuscript deals with an interesting topic and is based on sound methodology,

There are some points the authors might wish to address to further improve the manuscript.

Thank you very much for your review and valuable input. We will address each of your comments below.

1) Patients were divided in groups according to their PH and diastolic function. What was the respective rationale? Please specify.

Thank you for the question. Our rationale for dividing patients into two groups based on pulmonary hypertension (PH) status and diastolic function was based on the distinct physiological mechanisms through which these conditions influence pulmonary transit time (PTT) and pulmonary circulation parameters (PCP).

For pulmonary hypertension, right-heart catheterization represents the gold standard for diagnosis, and the elevated pulmonary artery pressure values determined by this method (>20 mmHg) officially define PH. Separating patients into PH and non-PH groups allowed us to assess whether PTT and PTTc could discriminate between patients with and without elevated pulmonary pressures using an established gold-standard reference. This grouping was therefore essential to evaluate the diagnostic utility of these CMR-derived parameters.

For diastolic dysfunction, transthoracic echocardiography remains the most reliable noninvasive method for grading diastolic function. Diastolic function is the ability of the myocardium to relax and adjust its compliance to allow adequate and pressure-efficient ventricular filling during diastole. It is essential for maintaining optimal hemodynamic stability; therefore, it affects PCP in another way. Grouping patients by diastolic dysfunction grade allowed us to examine whether PCP and related measurements reflect these known pathophysiological differences.

Our approach enabled us to study the link between CMR-derived parameters and established echocardiographic markers across clinically relevant categories of dysfunction.

2) Please provide quality assurance measures for CMR parameters (inter- and intra-observer agreement)

Thank you for bringing this crucial point to our attention. While the automated algorithm for onset detection is deterministic (ensuring that signal intensity curve generation and onset identification are fully reproducible for any given ROI placement), variability in manual ROI placement contributes to measurement variability between observers and repeated measurements. Therefore, we randomly selected 10 cases (12% of the study population) and assessed the inter- and intra-observer reproducibility for pulmonary circulation parameters (PTB and PTT) using the intraclass correlation coefficient (ICC) (type C, two-way mixed-effects model). We added this description to the "statistical methods" section:

"Intra- and inter-observer reproducibility were assessed using the intraclass correlation coefficient (ICC) (type C, two-way mixed-effects model with 95% confidence intervals (CI) from ten randomly selected cases. These cases were analyzed by two readers, one of whom repeated the analysis two weeks apart. The repeatability was classified as poor (<0.5), fair (0.50 to 0.75), good (0.75 to 0.90), and excellent (0.90 to 1)."

Also, we report now in the "Pulmonary circulation parameters" subsection from the "results" section in the revised manuscript this information, i.e.,

"Pulmonary circulation parameters intra-observer reproducibility was excellent for both PTB (ICC 0.992, 95% CI 0.970-0.998) and PTT (ICC 0.988, 95% CI 0.953-0.997). Likewise, inter-observer reproducibility was also excellent for PTB (ICC 0.982, 95% CI 0.934-0.996) and PTT (ICC 0.969, 95% CI 0.883-0.992)."

3) Is the software and algorithms used in this project freely available/transferable to other centers/machines?

We thank the reviewer for the comment. The PTB and PTT analyses were performed using a custom Python script, which is not publicly available and not for commercial use. However, our approach is entirely transferable and can be reproduced on other systems and scanners by following the methodology described in the manuscript: motion correction of first-pass perfusion images, manual ROI placement in the RV and LV with automatic propagation across frames, extraction of signal intensity curves, onset detection, and calculation of PTB and PTT as the difference between LV and RV onset points (frame number or time, respectively). Our algorithm uses standard image processing techniques and readily available peak-detection methods in Python libraries. Centers with basic programming capabilities can implement this workflow using the detailed methodology provided.

4) What was the time delay between the different assessments (CMR, Echo, RHC)?

Thank you for the question. In the case of CMR and RHC, as stated in the discussion, the time between the two modalities was at most 8 days, which we consider essential, as hemodynamics can change significantly depending on factors such as the patient’s cardiac compensation, volume status, etc. As part of the statistical analysis, we also tested a subgroup with a maximum interval of only 3 days between the examinations; the results were comparable. Therefore, in the final version of the article, we kept the larger cohort with examinations performed within 8 days.

The mean time delay between CMR and RHC was 1.8 ± 2.1 days, and between CMR and ECHO, 4.7 ± 12.1 days. We updated the discussion section accordingly.

5) Table 1: gender: the n is missing

Thank you for this point and we apologize for the omission. We updated the table to include the number of available values for each parameter, and also specified the number of male patients.

6) Table 1: “TEE” might rather the “TTE”

Thank you for noticing this typo. The table has been updated accordingly.

7) Table 1: the number of decimals should be equal for all numbers. One decimal might be sufficient

Thank you for pointing this out. We updated the corresponding values in the tables to retain the proper number of significant digits.

8) Table 1: for consistency, please also provide TTE LVEF and TR Vmax or TRmaxPG

Thank you for your detailed point. Indeed, it improves consistency. Therefore, we updated Table 1 to include other TTE parameters, such as LVEF (%), TRPG (mmHg) (representing “TRmaxPG”), and sPAP (mmHg) as well..

9) Since you report a correlation with IVS, please provide the respective value in table 1

Thank you for this insightful point. In the "Pulmonary circulation parameters" subsection from the "results" section, we previously stated that “Other parameters correlating with PCP included the interventricular septum (IVS) from the tissue Doppler imaging (TDI).” This sentence meant the correlation of PCP with the TDI-derived peak systolic myocardial velocity S”, which was already shown in Table 1. To clarify this point, we updated the respective part of the results section to: “Other parameters correlating with PCP included the peak systolic myocardial velocity (S') of the interventricular septum (IVS) assessed by pulsed-wave tissue Doppler imaging (TDI).” In addition, since we listed the correlations in Table 2, we also included the correlation of this parameter in the table.

10) What parameters was the diagnosis of RODCM based on? Signs/Symptoms, NT-proBNP, Imaging markers??

Thank you for raising this relevant point. The definition of the RODCM throughout the literature is not standardized and lacks unification. In our article, we included patients with newly diagnosed dilated cardiomyopathy with heart failure symptoms appearing in the last six months, referred to our center based on the results from other hospitals. This is the most common definition. We updated the introduction section accordingly to include this definition.

Review Comments to the Author

Reviewer #2:

General comments

The manuscript presents a retrospective cohort study analyzing pulmonary circulation parameters (Pulmonary Transit Beats [PTB], Pulmonary Transit Time [PTT], and corrected PTT [PTTc]) derived from cardiovascular magnetic resonance (CMR) imaging in comparison with right heart catheterization (RHC) and echocardiographic parameters, in 84 patients with recent-onset dilated cardiomyopathy. The authors demonstrate that PTT and PTTc correlate with systolic and diastolic cardiac function as well as hemodynamic indices from RHC, and that these metrics predict pulmonary hypertension (PH).

Thank you very much for your review and valuable input. In addition to your comments, we also addressed those from the editor and another reviewer to our updated manuscript.

Specific comments

1. Table 1 Formatting

- Please avoid reporting unnecessarily precise values, e.g., heart rate of 77.67 ± 17.08. Rounding should be applied where appropriate.

Thank you for your suggestion. We updated the tables to include only the proper number of significant digits..

- Add a column indicating the number of available data points for each parameter. If all values are complete, mention this briefly in the Methods section.

Thank you for mentioning this point. We added the information to the table where possible. In some cases, the data points werenot available for all patients. For example, E/A data were not available for all patients because patients with atrial fibrillation lack an “A” wave.

- Present mean and systolic pulmonary artery pressure as well as cardiac index (CI) in Table 1. The data for mPAP and CI are already presented on page 11, line 183.

Thank you for this valuable suggestion. We updated Table 1 and added information on systolic pulmonary artery pressure (sPAP) and CI.

- Define “pulmonary artery pressure index” in the Methods section.

Thank you very much for noticing this omission. In Table 1, “PAPi” is defined as the “pulmonary artery pressure index,” but instead it should have been “Pulmonary Artery Pulsatility Index.” We are a transplantation + LVAD center, and PAPi is a clinically validated hemodynamic parameter used to assess the RV function, which is crucial in patients before LVAD implantation. It is defined as: PAPI = (Pulmonary artery systolic pressure – Pulmonary artery diastolic pressure) / Right atrial pressure. The methods section and table have been updated accordingly. We apologize for this oversight.

- DPG refers to “diastolic pressure gradient,” not “Diastolic Pulmonary Artery Pressure” (page 11, line 174). DPG is the difference between pulmonary artery diastolic pressure and pulmonary capillary wedge pressure. Please provide this definition in the Methods.

Thank you for noticing this oversight. We have added the definition in the methods section.

2. Subgroup Division and Classification (Page 11, lines 184–187)

The numbers cited for PH subgroups do not add up to the total of 84 patients. Please verify and correct the classification.

- Remove the 28 “precapillary” entry; do not sum isolated precapillary patients with combined post- and precapillary PH.

- One patient remains unclassified; please clarify.

The correct breakdown should be:

- 37 patients without PH

- 47 patients with PH

--6 with precapillary PH.

--40 with postcapillary PH

---18 isolated postcapillary PH (IpcPH)

---22 combined post- and precapillary PH (CpcPH)

Thank you for raising this valuable comment. The PH and non-PH subgroups consisted of 47 and 37 patients, respectively, for a total of 84 patients. We believe the issue arises from a lack of clarity in the description we provided previously. Therefore, we revised the sentence for greater clarity. Including the “precapillary” category was indeed confusing; the original rationale was primarily statistical, but from a clinical perspective, it added ambiguity, so we removed it. The same applied to patients with postcapillary PH; therefore, we excluded this category as well. We retained only “isolated precapillary PH,” “isolated postcapillary PH,” and “combined pre- and postcapillary PH”, which also better reflects the guideline-based classification. Regarding the numbers, one patient had an mPAP of 21 mmHg, a PCWP of 13 mmHg, and a PVR of 1.8 WU, and thus could not be assigned to any category. In the literature, such cases are described as either “unclassified pulmonary hypertension” or “borderline/mildly elevated mPAP without a confirmed pre-capillary or post-capillary component.” The updated description is:

“The patients were divided into subgroups according to the signs of PH: 47 had signs of PH (PH group), and 37 had no signs (non-PH group). Among the 47 patients with PH, 6 had isolated precapillary PH, 18 had IpcPH, and 22 had CpcPH. In one particular case, the patient had elevated mPAP over 20 mmHg, therefore signs of PH, but neither elevated PCWP nor PVR; thus, the patient could not be assigned to any category.”

3. Please provide a more detailed characterization of the patients classified as having precapillary pulmonary hypertension. Specifically, did these individuals have underlying lung disease or other identifiable etiologies for precapillary PH? Alternatively, did they exhibit features typical of postcapillary PH, apart from the absence of PAWP elevation? Additionally, please report the average pulmonary artery wedge pressure (PAWP) for this subgroup.

Thank you for the opportunity to explain these details better. There were 6 patients classified as having precapillary PH in total. They showed a little bit higher LVEF than the whole population (27.1% ± 9.4 vs. 24.4% ± 9.3), comparable mPAP (23 mmHg ± 1.2 mmHg vs. 22.4 ± 8.6 mmHg) and lower PCWP (12.83 ± 0.9 mmHg vs. 14.3 ± 8.6 mmHg). The average pulmonary artery wedge pressure for subgroups with isolated postcapillary PH was even higher - 20.3 ± 5.1 mmHg. Additionally, from the 6 patients with precapillary PH, 3 were diagnosed with lung diseases at the time of the RHC; one patient was a smoker and drug user with high probability of underlying lung disease and in 2 cases the cause was unknown; therefore, idiopathic. Apart from the heart failure itself, they did not exhibit features typical of postcapillary PH.

We updated the results section to clarify this:

Of the six patients with precapillary PH, three were diagnosed with underlying lung disease at the time of the RHC; one patient was a smoker with a high likelihood of undetected lung pathology, and in two cases the cause remained unknown.”

4. Can you provide a table divided by patients with no PH, pcPH and prePH illustrating all CMR and RHC parameters?

Thank you for this suggestion. Yes, we can provide this table as a supplementary file. It illustrates the main CMR and RHC parameters for our cohort, and we hope it strengthens our findings.

5. Completeness of Table 1 (Page 11, line 188)

Table 1 does not present a complete list of evaluated parameters. Please ensure all relevant variables are included or adjust the related text accordingly.

Thank you for this suggestion. We updated the table with more evaluated parameters - including parameters such as LVEF (%), TRPG (mmHg) (representing “TRmaxPG”), sPAP (mmHg), mPAP (mmHg), and CI (L/min/m^2).

6. Discussion

- The first two discussion paragraphs are repetitive. Please streamline and avoid using nearly identical sentences.

- The discussion is brief. Please elaborate on the clinical advantage of assessing PTB, PTT and PTTc by CMR compared to standard echocardiographic evaluation of PH.

State whether PTT or PTTc can predict increased pulmonary artery wedge pressure (PAWP), as this would markedly increase the clinical utility of your results.

---

## [Decision Letter · Decision Letter 1]

7 Jan 2026

Comparison of pulmonary circulation parameters acquired by cardiovascular magnetic resonance with right heart catheterization and transthoracic echocardiography in patients with recent-onset dilated cardiomyopathy

PONE-D-25-33529R1

Dear Dr. Panovský,

We’re pleased to inform you that your manuscript has been judged scientifically suitable for publication and will be formally accepted for publication once it meets all outstanding technical requirements.

Kind regards,

Wolfgang Rudolf Bauer, M.D., Ph.D.

Academic Editor

PLOS One

Additional Editor Comments (optional):

Reviewers' comments:

Reviewer's Responses to Questions

**Comments to the Author**

Reviewer #2: All comments have been addressed

2. Is the manuscript technically sound, and do the data support the conclusions?

Reviewer #2: Yes

3. Has the statistical analysis been performed appropriately and rigorously?

Reviewer #2: Yes

4. Have the authors made all data underlying the findings in their manuscript fully available?

Reviewer #2: Yes

5. Is the manuscript presented in an intelligible fashion and written in standard English?

Reviewer #2: Yes

Reviewer #2: Thank you for the thorough revision and happy new Year!

All my comments were addressed. I have only one further recommendation. The column “number” in table 1 illustrates the total number of participants to acknowledge the number of missings. The lign “ male gender” however shows the number of male participants (N=64). I doubt that the gender was unknown in 20 individuals. Please correct the number.

**Do you want your identity to be public for this peer review?** For information about this choice, including consent withdrawal, please see our Privacy Policy

Reviewer #2: No

---

## [Editor Report · Acceptance letter]

27 Oct 2025

PONE-D-25-33529R1

PLOS One

Dear Dr. Panovský,

I'm pleased to inform you that your manuscript has been deemed suitable for publication in PLOS One. Congratulations! Your manuscript is now being handed over to our production team.

Kind regards,

on behalf of

Prof. Wolfgang Rudolf Bauer

Academic Editor

PLOS One